# Supplementation with >Your< Iron Syrup Corrects Iron Status in a Mouse Model of Diet-Induced Iron Deficiency

**DOI:** 10.3390/biology10050357

**Published:** 2021-04-22

**Authors:** Tatjana Pirman, Ajda Lenardič, Alenka Nemec Svete, Simon Horvat

**Affiliations:** 1Department of Animal Science, Biotechnical Faculty, University of Ljubljana, Groblje 3, 1230 Domžale, Slovenia; lenardic.ajda@gmail.com (A.L.); simon.horvat@bf.uni-lj.si (S.H.); 2Small Animal Clinic, Veterinary Faculty, University of Ljubljana, Gerbičeva 60, 1000 Ljubljana, Slovenia; alenka.nemecsvete@vf.uni-lj.si

**Keywords:** nutritional iron deficiency, oral supplementation, mice, >Your< Iron Syrup, Fe-sulphate

## Abstract

**Simple Summary:**

Iron deficiency remains a major global public health problem. When not properly addressed, it can deteriorate to anaemia. Iron deficiency with or without anaemia is associated with increased morbidity and mortality as well as with perturbed cognitive, behavioural, and motor development. Orally supplemented iron usually represents a first-line intervention to combat iron deficiency, being readily available, convenient, and effective. However, some oral formulations are poorly absorbed and can exhibit prominent side effects such as intestinal inflammation, pain, and nausea. We here examined the effects of >Your< Iron Syrup, one of the novel liquid iron-containing food supplements based on the microencapsulation technology in a mouse model of diet-induced iron deficiency. Several biomarkers of iron status improved significantly upon oral supplementation with >Your< Iron Syrup with no associated inflammatory response. Our results indicate that >Your< Iron Syrup was efficient in correcting iron deficiency and was well tolerated in our mouse model.

**Abstract:**

The objective of this study was to compare the effects of >Your< Iron Syrup, a novel oral liquid iron-containing food supplement, with the commonly prescribed iron sulphate (Fe-sulphate) in a mouse model of diet-induced iron deficiency. Standard inbred BALB/cOlaHsd mice were fed low-iron diet for 11 weeks to induce significant decrease in blood haemoglobin and haematocrit and were then supplemented by gavage with either >Your< Iron Syrup or Fe-sulphate for two weeks. In >Your< Iron Syrup group, several markers of iron deficiency, such as serum iron concentration, transferrin saturation and ferritin level were significantly improved in both female and male mice. Fe-sulphate induced similar responses, except that it did not significantly increase iron serum in females and serum ferritin in both sexes. Fe-sulphate significantly increased liver-iron content which >Your< Iron Syrup did not. Transcription of *Hamp* and selected inflammatory genes in the liver was comparable between the two supplementation groups and with the Control diet group. Some sex-specific effects were noted, which were more pronounced and less variable in males. In conclusion, >Your< Iron Syrup was efficient, comparable and in some parameters superior to Fe-sulphate in improving iron-related parameters without inducing a response of selected liver inflammation markers in a mouse model of diet-induced iron deficiency.

## 1. Introduction

Iron represents an essential micronutrient across all life forms, ranging from simple unicellular bacteria to complex multicellular organisms. It plays a crucial role in many biochemical reactions and is central to oxidative phosphorylation in the respiratory chain. In mammalian species, iron is an indispensable constituent of haemoglobin in erythrocytes, providing adequate production of red blood cells, which deliver oxygen to all body tissues [1]. Deficiency of iron leads to general deprivation of oxygen needed for energy metabolism with potentially serious consequences in different organs. Negative impacts on cognitive, behavioural, and motor development have been observed in humans as well as in animals [2] and correction of iron status generally has the potential to improve these outcomes [3].

Appropriate levels of iron are maintained in the body mainly by recycling of iron released from senescent erythrocytes, however, considerable amounts of iron must also be absorbed from the diet to replace the iron that is naturally lost from the body or additionally required for growth [4]. Absorption of iron is a tightly regulated process because its excessive amounts can be toxic. This is due to its redox chemistry and the propensity to damage cell components. Furthermore, no active excretion mechanism for iron exists in mammals—apart from natural daily losses through sweating, shedding of the gastrointestinal mucosal cells, and physiological blood loss, the secretion through urine has been demonstrated to be almost null [4,5]. According to the needs of the body, absorbed iron is either directly used for haemoglobin production, or is converted to hemosiderin complexes representing storage iron and transported to the liver, spleen, and the bone marrow. Iron sufficiency or deficiency is then signalled through the liver-produced hormone hepcidin, which acts as a master regulator of further iron absorption or its entry into the circulation [4].

In animals, anaemia due to iron insufficiency is very pronounced and studied in pigs and is traditionally managed by various forms of iron supplementation [6]. In humans, anaemia affects roughly one third of the world’s population of which approximately 50% is secondary to iron deficiency, being the single most prevalent, yet highly preventable nutritional disorder in the world [7,8]. Several complementary approaches can be implemented for prevention and management of nutritional iron deficiency, such as introduction of dietary modifications to increase consumption of iron-rich foods, universal fortification of staple foods with iron, and provision of iron in the form of oral supplements [7,8]. Among the latter, many different formulations are available, differing in efficacy and side effects profiles.

Efficacy of oral iron-based interventions critically depend on both bioavailability and tolerability of iron. When poorly absorbed, excess iron remains in the gut, leading to intestinal inflammation, manifesting at the level of gastrointestinal problems, including pain and nausea, which are well-known side effects of oral iron therapy [9]. Therefore, many approaches to improve iron formulations have been considered so far, with microencapsulation technology proven effective in satisfying both the bioavailability as well as the tolerability issues of iron salts. In suckling piglets, where iron deficiency anaemia represents the main malnutrition problem, encapsulated iron sulphate (Fe-sulphate) has proven to be more effective in prevention of anaemia than its non-encapsulated counterpart [10]. Similar results with ferric pyrophosphate were obtained in rats [11] and in humans [12,13,14,15,16]. Additionally, microencapsulated preparations of iron salts, such as Fe-sulphate, ferric chloride, and ferric citrate, which are most frequently used as supplements in animal [4,17] and human nutrition [18], were found to be less susceptible to inhibitors of iron absorption such as phytates, tannins and polyphenols [19,20], further endorsing their usefulness.

In the present study, we focused on the effects of >Your< Iron Syrup, one of the novel liquid iron-containing food supplements based on the microencapsulation technology, in a mouse model of dietary-induced iron deficiency. Animal models, in which iron levels in the body have been reduced through means of feeding the animals iron deficient diet, are particularly useful for studying iron bioavailability and the effects of different iron formulations on the correction of iron deficiency [21]. Side effects, such as inflammation, can be assessed at the organ level because animal models allow for comprehensive tissue analyses not possible in humans. We here present the effects of >Your< Iron Syrup as compared to Fe-sulphate on haemoglobin and haematocrit levels, on the iron content in blood and repository organs (liver, spleen), and analyse the differences in selected liver inflammatory response markers in a mouse model of dietary-induced iron deficiency.

## 2. Materials and Methods

### 2.1. Study Design

This is a simple two-way comparative study to test two iron-containing food supplements for their effects on iron status in a mouse model of diet-induced iron deficiency. A commonly prescribed Fe-sulphate was used as a positive control, a saline group as a negative control and the >Your< Iron Syrup as a tester group. Experimental plan is schematically displayed in Figure 1. The experimental unit was a single animal. We followed ARRIVE guidelines (https://arriveguidelines.org/arrive-guidelines, accessed on 12 April 2021) in the Material and Methods section to include all essential information in describing in vivo animal experiments.

### 2.2. Diets

Two diets were purchased from Research Diets (New Brunswick, NJ, USA): Open Standard Diet with 15 kcal% Fat—D11112201, used as iron-balanced diet (Control diet), and the same type of diet with no added iron—D15071503 (Iron deficient diet). Ingredients of the two diets as specified by a manufacturer Research Diets Inc. (New Brunswick, NJ, USA) are presented in Table 1. Before the experiment, samples of both diets were taken for the purpose of proximate analysis and mineral content. Proximate analysis was carried out in our laboratory by standard procedures [22]: dry matter on oven drying at 95–100 °C (AOAC Official method 934.01), crude protein by copper catalyst Kjeldahl method (AOAC Official method 984.13), crude fat to ether extract by extraction in petroleum ether (AOAC Official method 920.39), crude fibre by fritted glass crucible method (AOAC Official method 978.10) and crude ash (AOAC Official method 942.05). Macro and microelements (Ca, P, Mg, K, Na, Zn, Mn, Cu and Fe) were determined after ashing and preparation of an acid extract by atomic absorption spectrometry (1100B, PerkinElmer Inc., Waltham, MA, USA). Each sample was analysed in duplicate.

### 2.3. Experimental Products

>Your< Iron Syrup is a liquid iron-containing food supplement developed and manufactured by PharmaLinea Ltd. (Ljubljana, Slovenia, EU). Composition of >Your< Iron Syrup is of a proprietary nature but contains 14 mg of elemental iron in the form of branded micronised microencapsulated ferric iron (Qfer), 0.7 mg of vitamin B_6_, and 1.25 µg of vitamin B_12_ as active ingredients per 5 mL of the product. Fe-sulphate (cell-culture grade) was purchased from Sigma-Aldrich (St. Louis, MO, USA) and diluted in sterile and pyrogen free NaCl (0.9%, B. Braun, Melsungen, Germany). Duplicate samples of >Your< Iron Syrup were tested for the content of iron in our laboratory by atomic absorption spectrometry (1100B, PerkinElmer Inc., Waltham, MA, USA) after ashing of the product and preparation of acid extracts. 

### 2.4. Animals and Gavage Feeding Regime

All procedures with animals were performed according to current legislation on animal experimentation in Slovenia, which comply with the EU regulations regarding research on experimental animals. The protocol was approved by the Animal Ethics Committee of the Veterinary Administration of the Republic of Slovenia (project license number U34401-18/2019/4). The experiment was performed in the laboratory animals facility of the Department of Animal Science, Biotechnical Faculty, University of Ljubljana, Slovenia.

Eighty-four male and female (42 of each sex) BALB/cOlaHsd mice that were four weeks of age from Envigo-Harlan (Bresso, Italy) were housed in cages 11 cm (height) × 13 cm (width) × 30 cm (length) on a bedding of Lignocell (Mucedola Scobis Uno) and Rehofix (Mucedola Scobis Due G8). Animals had free access to drinking water and a pelleted diet. The room temperature was maintained between 21 and 22 °C and 45–55% humidity (checked and recorded daily), with light automatically regulated on a 12-h light/dark cycle starting at 6:00 a.m. Mice of the same sex were housed in groups of two to three per cage to follow widely recommended group housing enabling also the male mice to behave as social animals [23]. Based on our previous experience with this strain, and in accordance with some other comparative studies [24], males do not experience social stress or detrimental injurious fighting if housed together from early age of four weeks on, if a small portion of bedding material is transferred at cage cleaning, and if noise and disturbance are minimal. Cage enrichment and handling of mice was done by the tunnels made from clear polycarbonate plastic (two inches diameter, four inches long; Otto Environmental, Greenfield USA). This cage enrichment and methods for handling were recently recognised as having positive effects on stress reduction and on variability of results also in the BALB/cOlaHsd strain used here [25,26]. Animals were quarantined for the first 14 days. During this adaptation period, all animals received Control diet. Then, mice were randomly divided into two groups, one being fed Control diet and the other Iron deficient diet. We used the *randomize* command to assign mice to Control or Iron deficient diet groups and *combn* function to assign random combination of ear notch marks to each cage using the R programming package [27]. Equal number of male (21) and female (21) mice were assigned to each diet group. Animals received ad libitum Control or Iron deficient diet for 11 weeks. No animals were excluded based on the Hb or Ht measurements at the end of 11-week diet treatment although some mice developed mild or no iron deficiency in the Iron deficient diet group. All animals were used in subsequent two-week supplementation regime with either saline, Fe-sulphate, or >Your< Iron Syrup with the same number of male and female mice in each experimental group (Table 2). All products were administered by a 150 µL daily gavage. Fe-sulphate and >Your< Iron Syrup were appropriately diluted with saline (sterile and pyrogen free NaCl 0.9%, B. Braun Melsungen, Germany) to ensure the dosing of 1 mg of iron per kg body mass. Details on the experimental plan are given in Figure 1 and in Table 2. The leading investigator was aware of the group allocation at the different stages and coded cages and reagents such that the animal technician administering gavage was blind to the group treatments. Similarly, coded samples were sent to the laboratory to be tested blindly, for example for the iron determination in blood serum, etc. The researcher that analysed the data was given information regarding which data belong to a certain group but was not aware of which specific treatment a particular group received.

### 2.5. Haemoglobin and Haematocrit Determination

Haemoglobin concentrations and haematocrit levels were determined in a small drop of blood taken from a tail. Using a lancet, a small (1 mm) cut was made in a tail and a drop of blood was collected in a microcuvette, which was inserted in the measuring instrument (Hemo-Vet—EKF Diagnostics). At the end of the experiment, blood was collected from the body trunk and serum was prepared and frozen.

### 2.6. Iron Determination in Blood Serum and Liver

Serum iron concentration and total iron-binding capacity (TIBC) were measured on an automated biochemical analyser (RX Daytona; Randox, Crumlin, UK) using Randox reagent kits (serum iron, Cat. no.: SI3821; TIBC, Cat. no.: TI4064) according to the manufacturer’s recommendations. The measurement of serum iron concentration was based on the ferrozine method. Serum TIBC was measured using Randox TIBC colorimetric assay. Transferrin saturation was calculated from the concentration of iron in the serum normalised to the total iron-binding capacity of the available transferrin.

The amount of iron in the liver was determined after microwave digestion (Milestone microwave laboratory systems, Sorisole (BG), Italy). A sample of 0.25 g of liver, weighed to an accuracy of ±0.1 mg, was placed in a vessel and 7 mL of 65% HNO_3_ (suprapur) and 1 mL of 30% H_2_O_2_ was added. Microwave digestion of the samples was then performed using a two-step program: 10 min incubation to reach 200 °C and 1000 W, followed by a further 20 min digestion at 200 °C and 1000W. After completion of the program, the solution was cooled to room temperature and transferred to a marked flask. The concentration of iron in the clear solution was determined by atomic absorption spectrometry (1100B, PerkinElmer Inc., Waltham, MA, USA).

### 2.7. Ferritin Determination in Blood Serum and Liver

Serum ferritin level as well as the amount of ferritin in the liver homogenate was determined by the Mouse Ferritin ELISA Kit 157713 (Abcam, Cambridge, UK), essentially following manufacturer’s protocols. For the liver, the same part of liver was used in all group comparisons, namely, the right lateral lobe (*lobus dexter lateralis*). Frozen liver tissue samples were weighed, placed in homogenisation buffer, homogenised and supernatants were used for liver ferritin determination. The data for liver ferritin are expressed as μg ferritin per gram of liver tissue. Ferritin reacted with the anti-Ferritin antibodies adsorbed to the microtiter wells. Unbound proteins were removed by washing and horseradish peroxidase-conjugated anti-Ferritin antibodies were added to allow for the formation of immune complexes. These complexes were washed, and their quantity determined by absorbance at 450 nm. Standard curve was constructed from the standards supplied in the kit to be used for interpolation of test-sample ferritin quantity that was corrected for sample dilution.

### 2.8. qPCR Analyses of Hamp (Hepcidin Antimicrobial Peptide) and Selected Liver Inflammatory Marker Genes

Total RNA was isolated from the liver tissue samples using RNA PureLink RNA Mini Kit (Invitrogen, Waltham, MA, USA) according to manufacturer’s instructions. During the process of RNA isolation, DNA was removed from samples by Pure Link DNAse (Invitrogen, Waltham, MA, USA). cDNA was generated from 1 µg of RNA using High-Capacity cDNA Reverse Transcription Kit (Applied Biosystems Waltham, MA, USA) in a final volume of 10 µL. Each cDNA sample was diluted 1:10 and analysed by qPCR using PowerUp SYBR Green Master Mix (Applied Biosystems, Waltham, MA, USA) and Viaa7 machine (Applied Biosystems, Waltham, MA, USA). To assess the effect of supplementation on the key regulator of the entry of iron into the circulation, the *Hamp* (hepcidin antimicrobial peptide) gene was chosen. Mice also have a paralog *Hamp2* (hepcidin antimicrobial peptide 2) but it was shown previously that this gene had no effect on iron metabolism [28]. Selected liver inflammatory marker genes were used (*Crp, Il6, Saa1, Socs3*) as they were previously shown in a similar study to be informative [21].

Based on our preliminary experiments, *Gapdh* and *Actb* were chosen as reference genes because they showed stable expression across groups and uniform efficiency in dilution series of liver cDNA templates. All examined gene expression data were normalised to the geometric average of *Gapdh* and *Actb* dCt values. To compare expression data between groups, parametric test of dCt values were analysed by ANOVA using the SAS programming software SAS/STAT module (SAS Institute, Cary, NC, USA). Sequences of the primers used for qPCR are listed in Table 3.

### 2.9. Statistical Analysis

Results were analysed using general linear models (GLM) procedures of the SAS/STAT module (SAS Institute Inc., Cary, NC, USA), the differences being determined by a Tukey–Kramer multiple comparison test, taking into consideration the gavage as the main effect, separately for male and female mice. Least square means (LSM) are shown in the results, and the dispersion was expressed as the standard error of the mean (SEM). Statistical significance was considered at *p* ≤ 0.05. The difference in haemoglobin and haematocrit level before gavage treatment between the Control and Iron deficient diet groups (Figure 2 and Figure 3) were analysed by the Student *t*-test taking into consideration the diet as the only effect, separately for male and female mice.

## 3. Results

### 3.1. Chemical Composition of the Diets and of >Your< Iron Syrup

Results of proximate analysis of the Control and Iron deficient diets are shown in Table 4. As expected according to specifications, considerable differences between the two diets were found only in the content of iron. However, the amount of iron in the Iron deficient diet exceeded the specified concentration almost 5-fold. On the contrary, analysis of the amount of iron in >Your< Iron Syrup corresponded to the specified amount (see Section 2.2). 

### 3.2. Food Intake, Growth Traits and Organ Weights

Animals adapted well to the experimental conditions and no significant differences were found in food intake between mice on the Iron deficient and Control diet. Upon completion of the experiment, the average final body mass or total diet consumed did not differ significantly among groups of the same sex regardless of the diet type. Additionally, no differences in behaviour or health-pain assessment were detected between mice in the Iron deficient and Control diet groups according to regular inspections to recognise pain and assess its severity by the mouse grimace scale method [31,32]. The significant differences between males and females were in parameters known to be sexually dimorphic, such as body mass (*p* < 0.0001), amount of consumed diet (*p* = 0.0189), mass of liver (*p* < 0.0001), percentage liver per body mass (*p* < 0.0001) and percentage spleen per body mass (*p* < 0.0001), but not in the mass of spleen (*p* = 0.1386) [33,34]. Results are presented in Supplemental Appendix A.

### 3.3. Haemoglobin and Haematocrit Level

Haemoglobin and haematocrit measurements were done once per week for 11 weeks before starting with gavage to assess the degree of elicited iron deficiency in mice based on the changes in these parameters. The results in Figure 2 and Figure 3 show that at the end of 11-week dietary intervention, mice in Iron deficient diet group had significantly lower haemoglobin and haematocrit levels as compared to mice in the Control group. The reductions were more pronounced in male (*p* < 0.0001 and *p* < 0.00001) as compared to female mice (*p* = 0.0040 and *p* = 0.0117) in haemoglobin and haematocrit levels, respectively. These results indicate that feeding mice with Iron deficient diet for 11 weeks resulted in nutritional iron deficiency. 

To evaluate if Fe-sulphate and >Your< Iron Syrup could ameliorate or reverse the diet-induced iron deficiency, gavage supplementation was performed daily for 14 days according to the gavage plan (Table 2). The improvement of both haemoglobin and haematocrit levels in both Fe-sulphate and >Your< Iron Syrup groups is evident (Figure 4 and Figure 5) as both iron supplementations increased Hb and Ht levels in iron deficient males to the levels determined in Control diet groups. However, differences have just failed to reach the significance threshold of 0.05 in males, being for example at *p* = 0.0553 for haemoglobin and *p* = 0.0776 for haematocrit between >Your< Iron Syrup and Saline iron deficient groups. No significant differences were detected in female groups on Iron deficient diet, although a similar trend of improving Hb and Ht was also observed for both >Your< Iron Syrup and Fe-sulphate (Supplemental Appendix A, Appendix A).

### 3.4. Iron Levels in Serum and Liver

Following gavage of mice on Iron deficient diet, blood serum iron concentration significantly increased in male mice of the Fe-sulphate group and >Your< Iron Syrup group as compared to the Iron deficient Saline group (Table 5, left columns). In females, >Your< Iron Syrup also significantly increased serum iron concentration compared to the Iron deficient Saline group, but Fe-sulphate failed to reach significance threshold (*p* = 0.1709). Moreover, both iron supplementations resulted in serum iron concentrations that were not significantly different from mice receiving Control diets. Similar positive effects of both iron supplementations were determined for serum transferrin saturation as well (Figure 6). Fe-sulphate and >Your< Iron Syrup significantly increased transferrin saturation compared to the Iron deficient Saline groups in both sexes and reached similar and nonsignificantly different levels from the Control_saline group.

In the liver, effects of iron supplementations were not as pronounced as in serum. Neither Fe-sulphate nor >Your< Iron Syrup significantly increased non-heme hepatic iron, with the exception of Fe-sulphate in the female Iron deficient dietary group (Table 5, right columns). There, Fe-sulphate caused significant increase in iron levels as compared to the Iron deficient Saline group. This increased hepatic iron level was comparable to and not significantly different from the hepatic iron level in the Control_saline group. 

In general, two-week gavage with either Fe-sulphate or >Your< Iron Syrup restored iron levels and transferrin saturation in serum back to the levels found in the Control_saline diet group, but longer supplementation might have been required to also significantly increase the hepatic iron levels.

### 3.5. Ferritin Level in Serum and Liver

Serum ferritin concentration was significantly higher in >Your< Iron Syrup groups, both for male (*p* = 0.0230) and female (*p* = 0.0227) mice, as compared to Iron deficient Saline group. The Fe-sulphate group showed a trend toward increase in serum ferritin concentration as compared to the Iron deficient Saline group, but this increase was not significant (Figure 7). For liver ferritin levels (Figure 8), however, both iron supplementations led to significant increases in both sexes (*p* < 0.05) as compared to the Iron deficient Saline groups.

### 3.6. mRNA Expression of Hamp and Inflammatory Markers in the Liver

All relative expression data were calculated separately for each sex in order to analyse potential sex-specific effects. For the *Hamp* mRNA expression, some groups differed significantly and some differences between sexes were noted (Figure 9). In Fe-sulphate supplemented males on the Iron deficient diet, *Hamp* expression was increased in comparison with the Iron deficient Saline group, but not with other groups. Expression of *Hamp* in >Your< Iron Syrup group fed the Iron deficient diet was not statistically different from the Saline groups fed either the Iron deficient or the Control diet. In females, both >Your< Iron Syrup and Fe-sulphate showed a significant increase in *Hamp* mRNA levels as compared to the Saline group on the Iron deficient diet, but not compared to the Saline group maintained on the Control diet. There were no significant differences in *Hamp* expression between both iron supplementation groups on Iron deficient diet.

To examine the potential of iron supplementations to elicit an inflammatory response, expression of four genes, *Crp*, *Saa1, Il6*, and *Soc3* that were found informative in a similar experimental design to our study [21], were assayed by qPCR in the liver. As shown in Appendix A, neither Fe-sulphate nor >Your< Iron Syrup supplementations mounted an inflammatory response when expression of the four marker genes was compared to the mice receiving Control diet. For the *Saa1*, there were no significant statistical differences between iron deficient groups, but the iron deficient saline group did have significantly increased expression in comparison with the Control diet group. Sex specific effects were found significant for *Crp* where male groups had increased expression compared to females and for *Saa1*, where female groups had increased expression compared to male groups.

## 4. Discussion

Oral iron supplementation is a first line choice in individuals with iron deficiency and iron deficiency anaemia. Orally administered iron is convenient, inexpensive, and generally effective. Fe-sulphate remains to be the most frequently used preparation [19] but has a rather undesirable tolerability profile. Therefore, there is constant search for new oral preparations of iron with effectiveness comparable to Fe-sulphate but distinguished from it by better tolerability and side effect profile [1]. Here we tested >Your< Iron Syrup, a source of branded micronised, microencapsulated trivalent iron to examine its efficiency and side effect profile in a mouse model of diet-induced iron deficiency.

### 4.1. Efficacy of Iron Deficient Diet in Inducing Iron Deficiency in Mice

Inducing anaemia in standard inbred wild-type mice using iron-deficient diet as used in our study is not very efficient. First, mouse strains vary widely in iron-related parameters, such as the level of iron in the serum [35]. For example, a study comparing 32 mouse strains (Mouse phenome database No. 24426) revealed a wide range of values between strains from the lowest found in the C57BL/6J (~160 mMol/mL) to close to 400 mMol/mL in FVB/NJ with large variation within strains and between sexes. We here chose a standard strain BALB/cOlaHsd that exhibits low to moderate serum iron levels and has previously been used in diet-induced iron deficiency experiments [36]. Although we placed the mice on an iron-deficient diet soon after weaning, when they are supposed to be more sensitive to the development of anaemia [37], it took 11 weeks for Hb and Ht levels to drop for only approximately 10–15% (Figure 2). One explanation could lie in the choice of BALB/cOlaHsd substrain that could be more resistant to diet induced anaemia than other standard inbred strains. The C57BL/6J strain that is even less iron-laden than substrains of BALB/c is commonly used in such experiments. However, even with the C57BL/6J strain, mild anaemia developed very slowly and even not in all animals as reported previously [21]. One other plausible factor for slow and moderate development of iron deficiency is the fact that diet seems to be difficult to make with close to zero iron content. We used a defined purified diet D15071503 from Research Diets that was estimated by the company to contain around 3 mg Fe/kg diet. However, our own chemical composition analyses (Table 4) revealed that the diet contained nearly 5-fold more iron (14.8 mg Fe/kg diet), which is only around 3-fold lower concentration of iron than the one contained in the Control diet. As mice in the Iron deficient diet groups did not have a significantly different food intake as compared to the Control diet mice (Supplemental Appendix A), increased food intake could not play a major role to compensate for iron deficiency in the diet. It is possible that in our Iron deficient diet groups intestinal absorption of iron improved, an assumption supported by measurements of hepcidin mRNA transcription levels. Additionally, iron could have been recycled by increased coprophagy, which could be examined in future experiments in more detail with this model. It is a known fact that most of the iron entering blood plasma comes from recycling, so it is likely that lower amount of iron in our diet was even more efficiently recycled to resist reaching iron deficiency state in our mouse model [38].

For our experiment, we used the entire population of mice maintained on the Iron deficient diet rather than selecting out only the animals with significant reductions in Hb and Ht levels. Therefore, our experimental population was very heterogenous and consisted of lower to mildly iron-deficient animals. In the future, it may be more efficient to start out with a very large group of mice maintained on Iron deficient diet and select animals that develop at least a moderate level of anaemia to achieve a more uniform experimental population. All the above-mentioned limitations most likely contributed to the fact that the mean differences before and after iron supplementations were relatively small and often with large standard deviations. However, we were still able to demonstrate many significant improvements in iron-related parameters upon >Your< Iron Syrup supplementation that was comparable or in some instances even superior to the standard orally supplemented Fe-sulphate. 

### 4.2. Effects of Fe-Sulphate and >Your< Iron Syrup

Our prolonged Iron-deficient diet feeding regime in BALB/cOlaHsd mice managed to induce mild iron deficiency as assessed by small, but statistically significant decreases in Hb and Ht measurements (Figure 2 and Figure 3). Following a two-week oral administration of >Your< Iron Syrup and Fe-sulphate, a majority of iron-related parameters improved in a similar way between the two oral formulations, especially when compared to the saline-treated group. Both preparations increased iron concentrations in the serum of both sexes to a similar and relatively large extent (~1.5 to 2-fold) and only Fe-sulphate failed to reach significant threshold in females. Similar correctional effect was found for transferrin saturation in blood serum. Moreover, both oral iron supplementations in Iron deficient groups reached levels of serum iron and transferrin saturation that were comparable to and not significantly different from the Control diet group. As transferrin is the main iron binding protein in the blood, the fact that Fe-sulphate as well as >Your< Iron Syrup significantly increased its saturation in both sexes as compared to the Iron deficient Saline group strongly indicates a rescue of iron deficiency. Considering that transferrin saturation also reached levels of the Control diet groups further confirms this. Similar positive effects of both oral iron supplementations were found for ferritin concentrations in the serum and the liver as well. Consistently higher values in serum ferritin concentrations were obtained for >Your< Iron Syrup than for Fe-sulphate, although statistical significance was not reached due to high within group variability [39]. Both transferrin saturation and serum ferritin levels were found as informative markers in different underlying iron deficiencies in humans [40,41]. The fact that improvements in essentially all parameters in >Your< Iron Syrup supplementation group were similar to the Fe-sulphate group used as a positive control, confirms that >Your< Iron Syrup is efficient in correcting nutritional iron deficiency in our mouse model.

### 4.3. Expression of Hepcidin (Hamp) and Inflammatory Marker mRNAs in the Liver

Hepcidin has been shown to be one of the main regulators of iron metabolism [42]. Various studies in humans and in mice have demonstrated that hepcidin inhibits absorption of iron from dietary sources in intestine as well as affects placental iron transport, and recycling in macrophages, which decreases the delivery of iron to developing erythrocytes [43]. If *Hamp* was overexpressed, mice died shortly after birth from severe iron deficiency. In contrast, in the mouse *Hamp* knockout model, inhibitory role was removed resulting in iron overload resembling human hemochromatosis [44]. Accordingly, if mice were overloaded with iron, either by diet or genetically, hepcidin mRNA expression was increased [45], but in mice with anaemia, hepcidin mRNA was decreased [46]. Results of this latter study are in line with our data in that mice fed the Iron deficient diet without iron supplementation (Saline) showed the lowest mRNA expression of *Hamp*. Fe-sulphate supplemented males on Iron deficient diet had significantly increased *Hamp* transcription compared to both Control saline groups. This could be seen as a negative effect of Fe-sulphate not observed in >Your< Iron Syrup group. Similar significant increase of *Hamp* transcription due to Fe-sulphate supplementation was observed in the study of Asperti et al. [21], the effect ascribed to the upregulation of inflammatory markers upon Fe-sulphate administration. Since the expression of hepcidin in liver can be induced either by iron overload or by inflammation, mRNA levels of inflammatory markers *Crp*, *Il6, Saa1, Socs3* were analysed. No significant increase in the transcription of any of these genes was observed in either supplementation group or in either sex when compared to Control diet groups (Supplemental Appendix A). Sex specific effects were found for *Crp* and *Saa1*—for the later gene, similar sex-bias, although for a different trait, was shown in the same mouse strain as used in our study [47]. Overall, based on the comparison to the Control diet group, we can conclude that >Your< Iron Syrup did not induce an inflammatory response as assessed by qPCR analysis of liver *Crp*, *Il6*, *Saa1*, and *Socs3*.

### 4.4. Sex Differences

Many traits and physiological systems exhibit differences between sexes [48]. In preclinical animal model research, investigators tend to use one sex only. More often male rodents are used either because researchers assume that there are no differences between sexes, or the phenotype is expressed better, or due to practical reasons—due to harem mating practices there are usually extra males available in rodent colonies with females reserved for breeding. Such one-sex research designs hinder and delay scientific discoveries especially in understanding of mechanisms for these differences and in developing targeted sex-based treatments. Some funding agencies such as the US National Institutes of Health (NIH) have recently adopted policies that require applicants to report their plans for the balance of male and female cells and animals in preclinical studies and all future applications [49]. We hence included both sexes and revealed some differences between sexes in baseline parameters as well as in response to iron supplementation. Following an 11-week iron deficient diet treatment, males developed more pronounced haematological changes than females as assessed by Hb and Ht measurements (Figure 2 and Figure 3). A response to oral supplementation with >Your< Iron Syrup was clearly more pronounced for Hb and Ht in males (Figure 4 and Figure 5) than in females (Supplemental Appendix A). A study by Hahn et al. [47] found that various tissues in different mouse strains exhibit sexual dimorphism in terms of iron content. For example, they found that liver stores of iron can be increased 2–3-fold in females versus males. A substantial increase of liver iron level in the females compared to males was also observed in our experiment (Table 5). It is possible that increase in hepatic iron in females made them more resistant to feeding Iron deficient diet. In contrast, a response to >Your< Iron Syrup was much more pronounced in female than male mice for ferritin levels—for example, in the liver, the difference between females and males was two-fold (194 µg/g liver in females versus 86 µg/g liver in males). In several parameters, such as in serum and hepatic iron and ferritin levels, variability was larger in females than in males, possibly due to the effect of oestrous cycle in females that can affect certain metabolic traits [48]. Future studies should be designed such that the mechanism of sex differences can be studied in more detail to identify optimal sex-based interventions for iron deficiency.

### 4.5. Limitations

Our study, as most animal model studies, has some limitations. We already mentioned a relatively mild effect of iron deficient diet feeding on inducing anaemia which should be addressed in future experiment by ensuring that the iron deficient diet indeed contains close to zero amount of iron. Our results also pertain only to the BALB/cOlaHSD mouse substrain, so the positive effects found for >Your< Iron Syrup may not be detected also in other genetic background. Other iron supplements than Fe-sulphate that was used here as a comparator and a positive control group could also be used, especially as Fe-sulphate has been shown to cause gastro-intestinal and other side effects such as inflammatory response [21]. Another problem with using rodent models for mineral studies is rodents’ tendency for coprophagy. They use this as a way to recycle minerals including iron, which may also have a dramatic impact on the results of a nutritional study [50]. There are methods to prevent rodents from practicing coprophagy, for example by rearing them on metal grids instead of bedding, but in our hands this introduces great stress to the animals and is likely to result in nonphysiological and more variable responses. Finally, rodent models have significant differences to humans in iron metabolism which makes extrapolation of our results to humans questionable. Nevertheless, a relatively good response obtained after only two weeks of oral supplementation in our study is encouraging, particularly considering that most analysed parameters of iron status have improved in both sexes with no side effects noted. In line with this is a recently published parallel double-blind randomised clinical study using >Your< Iron Syrup in iron-deficient children, showing similar efficacy and safety effects as found in our animal study.

## 5. Conclusions

We demonstrate that a standard, non-gene modified BALB/cOlaHsd mouse strain can be used to induce nutritional iron deficiency by feeding low iron content diet to study supplementation effects of different oral iron formulations. Several markers of iron deficiency, such as serum iron, transferrin saturation, and hepatic ferritin level were significantly improved by administration of >Your< Iron Syrup and a positive control Fe-sulphate in both sexes. Some other parameters (i.e., serum ferritin concentrations) increased significantly in >Your< Iron Syrup but not in Fe-sulphate group, though the trend in the latter was positive as well. Both supplements similarly improved Hb and Ht values following a two-week gavage. Compared to the negative control group (the Iron deficient Saline), both >Your< Iron Syrup and Fe-sulphate activated *Hamp* transcription, an inhibitor of iron absorption, but only to the level of the *Hamp* expression in the group receiving normal quantity of iron in the diet (the Control_saline group). Some sex-specific effects were noted; generally, the positive effects were more pronounced and less variable in males than in females. Our results show that >Your< Iron Syrup is efficient, comparable and, in some parameters, even better than Fe-sulphate in improving iron-related parameters without causing an inflammatory response in our preclinical mouse model of nutritional iron deficiency.

## Figures and Tables

**Figure 1 biology-10-00357-f001:**
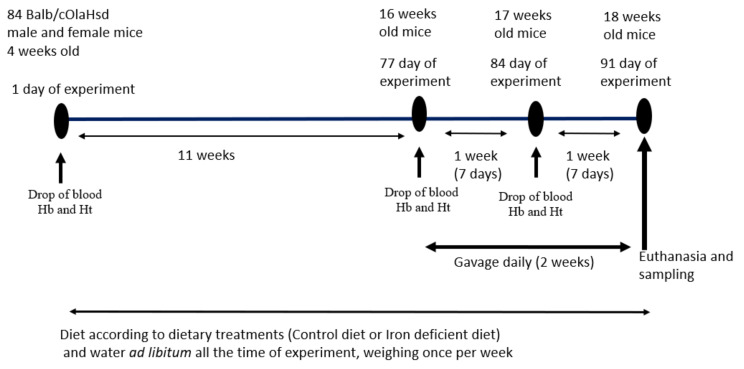
Experimental plan and gavage feeding regime. Drop of blood was taken on designated time points for Hb (haemoglobin) and Ht (haematocrit) determination.

**Figure 2 biology-10-00357-f002:**
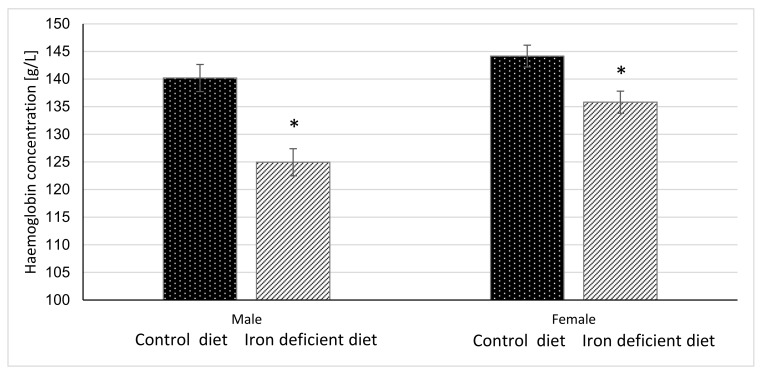
Haemoglobin concentration in mice before gavage feeding (average ± SEM). * Significant difference between Control and Iron deficient diet groups of the same sex: male *p* < 0.0001, female *p* = 0.0040.

**Figure 3 biology-10-00357-f003:**
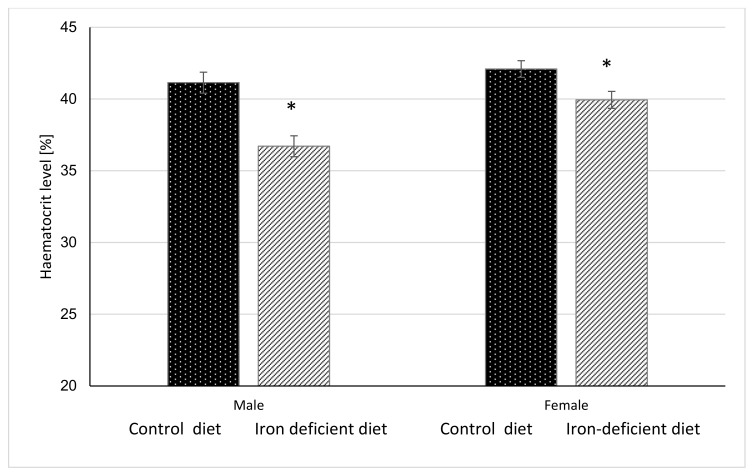
Haematocrit level in mice before gavage feeding (average ± SEM). * Significant difference between Control and Iron deficient diet groups of the same sex: male *p* < 0.0001, female *p* = 0.0117.

**Figure 4 biology-10-00357-f004:**
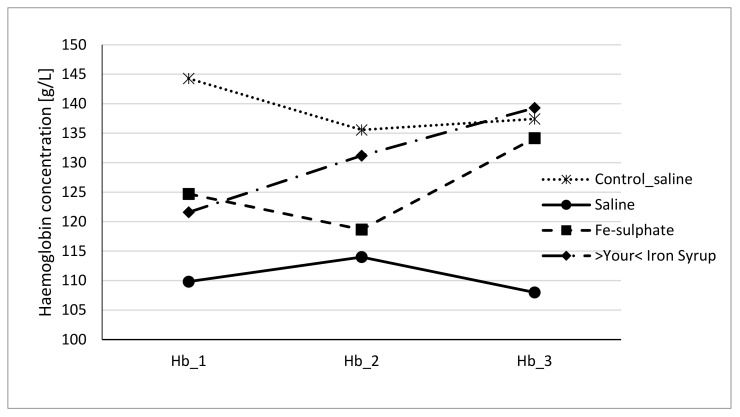
Haemoglobin concentration of male mice at the beginning of the gavage feeding (Hb_1), after one week (Hb_2) and at the end of two-week gavage feeding regime (Hb_3).

**Figure 5 biology-10-00357-f005:**
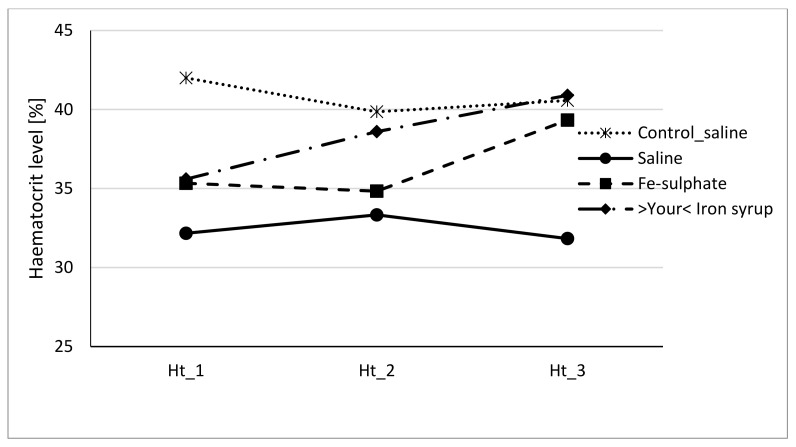
Haematocrit level of male mice at the beginning of the gavage feeding (Hb_1), after one week (Hb_2) and at the end of two-week gavage feeding regime (Hb_3).

**Figure 6 biology-10-00357-f006:**
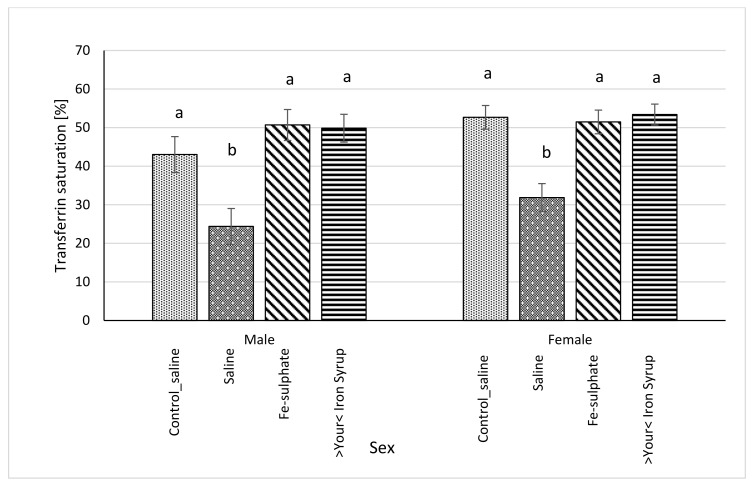
Transferrin saturation (%) calculated from serum concentrations of iron (µg/dL) and TIBC (µg/dL) (average ± SEM). ^ab^ Columns without the same superscript differ significantly between the groups inside the sex (*p* ≤ 0.05).

**Figure 7 biology-10-00357-f007:**
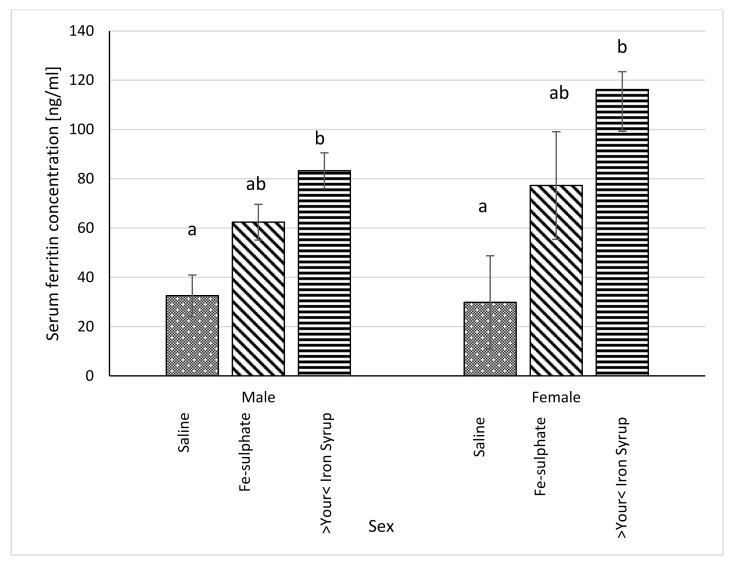
Ferritin concentration in blood serum in Iron deficient diet groups (ng/mL plasma) (average ± SEM). ^ab^ Columns without the same superscript differ significantly between the groups inside the sex (*p* ≤ 0.05).

**Figure 8 biology-10-00357-f008:**
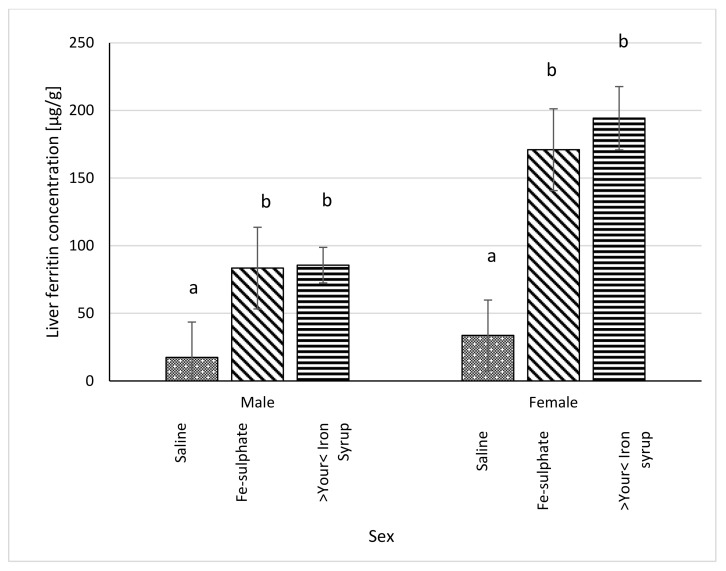
Ferritin concentration in liver (µg/g liver) of Iron deficient diet groups (average ± SEM). ^ab^ Columns without the same superscript differ significantly between the groups inside the sex (*p* ≤ 0.05).

**Figure 9 biology-10-00357-f009:**
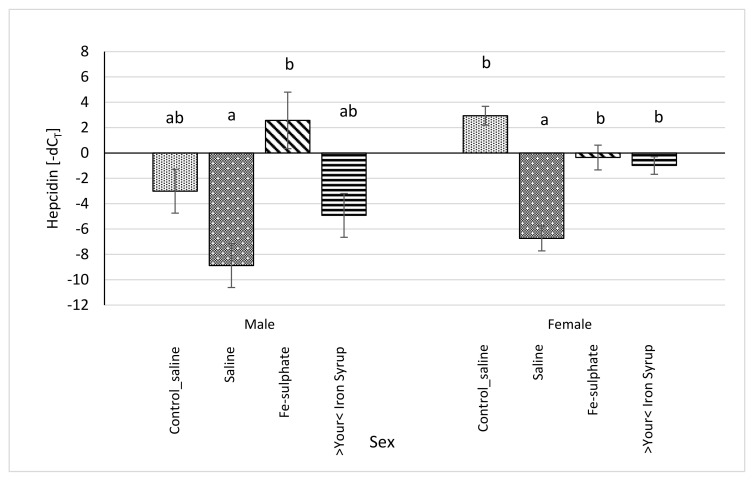
Relative hepcidin (*Hamp*) mRNA expression (shown as -dC_T_ values) of Iron deficient diet groups (average ± SEM) compared to the Control_saline group. ^ab^ Columns without the same superscript differ significantly between the groups inside the sex (*p* ≤ 0.05).

**Table 1 biology-10-00357-t001:** Ingredients of the diets (g) (reported ingredients); Open Standard Diets with 15 kcal% Fat (Research Diets).

	Control Diet (D11112201)	Iron Deficient Diet(D15071503)
Casein	200	200
L-Cystine	3	3
Corn Starch	381	381
Maltodextrin 10	110	110
Dextrose	150	150
Cellulose, BW200	75	75
Inulin	25	25
Soybean Oil	70	70
Mineral Mix S10026 ^1^	10	
Mineral Mix S18708 (no added iron) ^2^		10
Dicalcium Phosphate	13	
Dicalcium Phosphate, Chemically Pure		13
Calcium Carbonate	5.5	
Calcium Carbonate, Reagent Grade		5.5
Potassium Citrate, 1 H_2_O	16.5	16.5
Vitamin Mix V10001 ^3^	10	10
Choline Bitartrate	2	2
Yellow Dye #5, FD&C	0.025	
Blue Dye #1, FD&C	0.025	0.05
Total	1071.05	1071.05

^1^ Mineral mixture S10026 (10 mg/kg diet), (g/kg mineral mixture): Sodium Chloride 259 g, Magnesium Oxide, Heavy, DC USP 41.9 g, Magnesium Sulfate, Heptahydrate 257.6 g, Ammonium Molybdate Tetrahydrate 0.3 g, Chromium Potassium Sulfate 1.925 g, Copper Carbonate 1.05 g, Ferric Citrate 21 g, Manganese Carbonate Hydrate 12.25 g, Potassium Iodate 0.035 g, Sodium Fluoride 0.2 g, Sodium Selenite 0.035 g, Zinc Carbonate 5.6 g, Sucrose 399.105 g. ^2^ Mineral mixture S18708 (no added iron) (10 mg/kg diet), (g/kg mineral mixture): Sodium Chloride 259 g, Magnesium Oxide, Heavy, DC USP 41.9 g, Magnesium Sulfate, Heptahydrate 257.6 g, Chromium Potassium Sulfate 1.925 g, Copper Carbonate 1.05 g, Sodium Fluoride 0.2 g, Potassium Iodate 0.035 g Manganese Carbonate Hydrate 12.25 g, Ammonium Molybdate Tetrahydrate 0.3 g, Sodium Selenite 0.035 g, Zinc Carbonate 5.6 g, Sucrose 420.105 g. ^3^ Vitamin mixture V10001 (10 g/kg diet) (g/kg vitamin mixture): Vitamin A Acetate (500,000 IU/g) 0.8 g, Vitamin D3 (100,000 IU/g) 1 g, Vitamin E Acetate (500 IU/g) 10 g, Menadione Sodium Bisulfite (62.5% Menadione) 0.08 g, Biotin (1%) 2 g, Cyanocobalamin (0.1%) 1 g, Folic Acid 0.2 g, Nicotinic Acid 3 g, Calcium Pantothenate 1.6 g, Pyridoxine-HCl 0.7 g, Riboflavine 0.6 g, Thiamin HCl 0.6 g, Sucrose 978.42 g.

**Table 2 biology-10-00357-t002:** Composition of solutions used in gavage feeding.

Iron Deficient Diet Groups		Control Diet Groups	
Saline	Iron deficient diet12th and 13th week each day 150 µL 0.9% saline solution	Saline	Control diet12th and 13th week each day 150 µL 0.9% saline solution
Fe-sulphate	Iron deficient diet12th and 13th week each day 1 mg iron (from Ferrous sulphate)/kg body mass dissolved in 150 µL 0.9% saline solution	Fe-sulphate	Control diet12th and 13th week each day 1 mg iron (from Ferrous sulphate)/kg body mass dissolved in 150 µL 0.9% saline solution
>Your< Iron Syrup	Iron deficient diet12th and 13th week each day 1 mg iron (from >Your< Iron Syrup)/kg body mass—9 µL of >Your< Iron Syrup was mixed with 141 µL 0.9% saline solution	>Your< Iron Syrup	Control diet12th and 13th week each day 1 mg iron (from >Your< Iron Syrup)/kg body mass- 9 µL of Your Iron Syrup was mixed with 141 µL 0.9% saline solution

**Table 3 biology-10-00357-t003:** Primers used to quantitate mRNA expression of selected genes.

Target Gene	Primer Sequence	Amplicon Size (bp)	Primer Source
*Actb*	F: 5′-GTGACGTTGACATCCGTAAAGA-3′	245	PrimerBank [29] ID = 145966868c1
	R: 5′-GCCGGACTCATCGTACTCC -3′		
*Gapdh*	F: 5′-TCACCACCATGGAGAAGGC-3′	168	Giulietti et al. [30]
	R: 5′-GCTAAGCAGTTGGTGGTGCA-3′		
*Hamp*	F: 5′-AAGCAGGGCAGACATTGCGAT-3′	141	Asperti et al. [21]
	R: 5′-CAGGATGTGGCTCTAGGCTATGT-3′		
*Crp*	F: 5′-GCTACTCTGGTGCCTTCTGATCA-3′	135	Asperti et al. [21]
	R: 5′-GGCTTCTTTGACTCTGCTTCCA-3′		
*Il6*	F: 5′-CTCTGCAAGAGACTTCCATCCAGT-3′	58	Asperti et al. [21]
	R: 5′-CGTGGTTGTCACCAGCATCA-3′		
*Saa1*	F: 5′-AGAGGACATGAGGACACCAT-3′	85	Asperti et al. [21]
	R: 5′-CAGGAGGTCTGTAGTAATTGG-3′		
*Socs3*	F: 5′-TTAAATGCCCTCTGTCCCAGG-3′	51	Asperti et al. [21]
	R: 5′-TGTTTGGCTCCTTGTGTGCC-3′		

**Table 4 biology-10-00357-t004:** Chemical compositions of the diets per kg of dry matter.

	Control Diet (D11112201)	Iron Deficient Diet (D15071503)
Dry matter (g/kg)	914.8	919.3
Crude fibre (g)	71.7	72.3
Crude fats (g)	69.5	69.4
Crude protein (g)	171.6	177.7
Crude ash (g)	34.8	35.5
Non-nitrogen extractives (g)	645.1	652.3
Minerals		
Na (g)	1.17	1.16
K (g)	6.92	6.31
Ca (g)	6.23	5.96
Mg (g)	0.60	0.61
P (g)	4.49	4.35
Fe (mg)	51.42	14.76
Zn (mg)	80.29	72.48
Cu (mg)	7.53	7.10
Mn (mg)	58.36	58.25

**Table 5 biology-10-00357-t005:** Level of iron (Fe) in blood serum and in liver separately for male and female mice (average ± SEM).

	Fe in Serum (µMol/L)		Fe in Liver (µg/g Liver)	
	Male	Female	Male	Female
Control_saline	31.93 ± 2.74 ^ab^	39.89 ± 2.99 ^ab^	124.34 ± 7.01 ^a^	158.37 ± 9.64 ^a^
Iron deficient diet groups				
Saline	21.03 ± 2.74 ^b^	32.41 ± 2.99 ^b^	55.54 ± 7.01 ^c^	68.96 ± 9.64 ^b^
Fe-sulphate	37.85 ± 2.56 ^a^	42.24 ± 2.99 ^ab^	89.80 ± 6.56 ^b^	139.06 ± 9.64 ^a^
Your Iron Syrup	39.27 ± 2.29 ^a^	44.40 ± 2.50 ^a^	63.74 ± 5.86 ^bc^	81.42 ± 8.06 ^b^

^abc^ Average values without the same superscript in the column differ significantly between the groups (*p* ≤ 0.05).

## Data Availability

Not applicable.

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
