# Peer review of "Supplementation with >Your< Iron Syrup Corrects Iron Status in a Mouse Model of Diet-Induced Iron Deficiency"

_biology, 2021, doi:10.3390/biology10050357_

Round 1

Reviewer 1 Report

"Following an 11-week iron deficient diet treatment, males developed more pronounced haematological changes than females as assessed by Hb and Ht measurements (Figures 2 and 3). A response to oral supplementation with >Your< Iron Syrup was clearly more pronounced for Hb and Ht in males (Figure 4 and 5) than in females (Supplemental Figures S1 and S2)." - These assertions are not valid; H and Ht were not compared with the appropriate statistical analysis (see the Methods - "The haemoglobin and haematocrit measurements were analysed by the Student t-test using SAS/STAT module taking into consideration the diet as the main effect, separately for male and female mice."

Author Response

Thank you for your recommendations for improving our manuscript.

In the attached file is our response and explanations of change we made in the manuscript.

Reviewer 2 Report

Dear authors,

Thank you for sharing your research with us! I really appreciate your manuscript, which is well prepared. Nevertheless, I have some concerns to improve your manuscript:

Introduction:

Nothing to add, I am fine with that.

Methods:

I was wondering about the organization of your methods. I expected a short overview of what was done, but you start with "Diets". This is confusing. This wasn't a randomized-controlled trial, right? Please see and consider CONSORT (or another checklist, if it is not suitable) for reporting trials.

What about sample-size calculation? What was the reason for choosing 84 mices? To proof effieciency of syrup, you need accurate sample-size calculation. Otherweise your trial is an exploratory trial/pilot trial gaining tendencies, which must be proven by further research.

So, please add sample-size calculation or change your wording results.

Results:

Nothing to add, I am fine with that. 

Discussion:

Discussion about limitations is too short. Transferability of mice results are always limited.

Conclusion:

See above. You are not able to draw firm conclusion without sample-size calculation and thorough statistical planning before onset. Furthermore, you are not able to draw any conclusions about syrup usage in humans.

Author Response

(The authors gave the same response as above.)

Reviewer 3 Report

This is an interesting study investigating a novel liquid based micro-encapsulated propriety iron formulation in mice with nutritionally induced iron deficiency anaemia (IDA).

The experiments have been well thought out, results clearly presented and the analysis robust. Well done checking iron levels in the "low iron" commercial chow - frustrating!- but valuable for other researchers to know.

The authors demonstrate non-inferiority c/w FeSO4 in terms of iron replenishment and selected inflammatory markers. Of course, one of the reasons they suggest FESO4 - a very inexpensive iron supplement- is not such a great supplement is that to may cause nausea, vomiting and other GI side effects- and their research in mice can not address this point. This should be clearly stated as a limitation.

Also, in several places the authors state that "Your" syrup corrects nutritional IDA - the authors must always specify "in our mouse model".

Other specific points

line 12 - what "chronic diseases", other than IDA, does ID cause?

line 35 - as the authors do not measure systemic markers of inflammation but only liver this should be specified - circulating inflammatory cytokine levels depend not only on production but systemic effects

lines 59-60 - by "inactive ferritin" do the authors mean hemosiderin?

line 66 - I think 2 billion people have anaemia, of which approx 50% is secondary to iron deficiency - if in fact two billion people have iron deficiency can the authors provide a reference that specifically supports this?

lines 97-98 - please specify site (liver, blood, systemic)  inflammatory markers and consider stating selected markers, limited markers etc - as there are many inflammatory markers and the authors use only a few

lines 185-190 - not sure what this is supposed to be? 

Re ferritin concentrations - how did the authors correct for cell counts, amount of tissue used when measuring ferritin?

in the section on qPCR analysis - the written section- inflammatory markers should be stated - not just shown in the accompanying table

lines 413 - please provide reference to support this

Discussion - section starting "Efficacy of ......" the discussion on models to induce anaemia in mice is irrelevant to this paper other than the nutritional models - I think this whole paragraph contains extraneous material and needs revising

line 495- correcting nutritional iron deficiency IN MICE - or in our mouse model

498 - hepcidin was first identified in 2000 and its role in iron metabolism known for decades - I am not sure this would be classified as "recently" anymore?

line 500 - after transport needs a comma - otherwise the authors are talking about recycling in macrophages in the placenta

line 557 - specify mouse

line 573- mouse model

Author Response

(The authors gave the same response as above.)

Round 2

Reviewer 2 Report

Thank you for revision of your manuscript. I'm fine with your changes and wish you all the best for publication of your research!

Reviewer 3 Report

The authors have addressed all the issues or questions that I had with the original submission.

One final minor point to consider - for Table One - "Table 1. Ingredients of the diets (g)" - were these the reported ingredients as stated on the label? or measured? I think it would be helpful to distinguish between reported ( and source) or measured where appropriate